# Effect of Nucleic Acid Screening Measures on COVID-19 Transmission in Cities of Different Scales and Assessment of Related Testing Resource Demands—Evidence from China

**DOI:** 10.3390/ijerph192013343

**Published:** 2022-10-16

**Authors:** Qian Gao, Wen-Peng Shang, Ming-Xia Jing

**Affiliations:** 1Department of Public Health, Shihezi University School of Medicine, Shihezi 832000, China; 2Shihezi City Center for Disease Control and Prevention, Eighth Division, Xinjiang Production and Construction Corps, Shihezi 832000, China

**Keywords:** COVID-19, nucleic acid screening, dynamical model of infectious disease, laboratory testing

## Abstract

Background: COVID-19 is in its epidemic period, and China is still facing the dual risks of import and domestic rebound. To better control the COVID-19 pandemic under the existing conditions, the focus of this study is to simulate the nucleic acid testing for different population size cities in China to influence the spread of COVID-19, assess the situation under different scenarios, the demand for the laboratory testing personnel, material resources, for the implementation of the nucleic acid screening measures, emergency supplies, and the configuration of human resources to provide decision-making basis. Methods: According to the transmission characteristics of COVID-19 and the current prevention and control strategies in China, four epidemic scenarios were assumed. Based on the constructed SVEAI_i_QHR model, the number of people infected with COVID-19 in cities with populations of 10 million, 5 million, and 500,000 was analyzed and predicted under the four scenarios, and the demand for laboratory testing resources was evaluated, respectively. Results: For large, medium, and small cities, whether full or regional nucleic acid screening can significantly reduce the epidemic prevention and control strategy of different scenarios laboratory testing resource demand difference is bigger, implement effective non-pharmaceutical interventions and regional nucleic acid screening measures to significantly reduce laboratory testing related resources demand, but will cause varying degrees of inspection staff shortages. Conclusion: There is still an urgent need for laboratory testing manpower in China to implement effective nucleic acid screening measures in the event of an outbreak. Cities or regions with different population sizes and levels of medical resources should flexibly implement prevention and control measures according to specific conditions after the outbreak, assess laboratory testing and human resource need as soon as possible, and prepare and allocate materials and personnel.

## 1. Introduction

As of 14 August 2022, the global novel coronavirus (COVID-19, new crown pneumonia) pandemic that broke out at the end of 2019 had 600 million confirmed cases and 6.45 million deaths worldwide [1]. The emergence of multiple novel coronavirus variants (such as Delta and Omicron) may lead to increased virus transmissibility and reduced vaccine effectiveness, making epidemic prevention and control more difficult [2]. The COVID-19 epidemic has not only caused a huge impact on global economic and social development, but also brought a huge threat to the physical and mental health of the population [3,4,5,6].

Under the circumstance that the international environment has not been fundamentally improved, China is facing greater pressure from overseas imports. Although China has adopted measures such as “7 + 3” isolation medical observation and health monitoring for inbound personnel [7], the current domestic aggregation of the epidemic still shows a trend of multiple points, frequent outbreaks, and wide-ranging spread. Since 2021, large-scale epidemics have occurred in Yunnan, Hangzhou, Chengdu, Shanghai, and Beijing [8,9,10,11,12]. To curb the epidemic, many countries have implemented non-pharmaceutical interventions (NPIs) such as lockdowns, mask-wearing, and social distancing [13], and the implementation of these NPIs has also changed people’s lifestyles. Now, people often wear masks when they go out and social distancing in public places. Additionally, studies have shown that social distancing of 1 m or more or the use of masks significantly reduces the incidence of COVID-19 infection [14].

Due to the increasing proportion of asymptomatic infection cases, routine nucleic acid screening in the population has become an effective and important prevention and control measure. Therefore, in the prevention and control of the COVID-19 epidemic, once a case occurs in various regions of China, large-scale nucleic acid screening is immediately organized to detect virus infection in time and curb the spread of the epidemic [15].

However, the cost of epidemic prevention brought about by the implementation of large-scale nucleic acid screening and other prevention and control measures has continued to rise, which has caused a huge impact on the medical and health systems of various countries [16]. Many countries have difficulty coping with the accompanying laboratory testing personnel and testing equipment. The demand for health resources such as health resources and reagents has surged, and the shortage of health resources is facing allocation problems [17]. Judging from the actual situation of the prevention and control of the COVID-19 epidemic, there are problems with insufficient laboratory testing and protective material reserves around the world [18]. Therefore, under the background of severe epidemic prevention and control situations, it is very important to assess the demands of laboratory testing resources and human resources as soon as possible to prepare for the allocation of manpower and material reserves.

Previous studies have shown that the SEIR model can well fit the development trend of the COVID-19 epidemic [19,20,21], but the global medical resource demands assessment research specifically for the emergency prevention and control of emerging infectious diseases such as COVID-19 is not yet available limited. A United States study used a transmission dynamics SEIR model to estimate medical demands in the COVID-19 pandemic and to assess the potential impact of a society-wide intervention [22]. Chinese researchers Wang Qing et al. [23] also used the improved SEIR model to research the dynamics of new crown transmission and the assessment of resource demands for prevention and control. Most of these studies focus on the assessment of clinical medical resources, and most of them only focus on the population size of a city, and there are fewer studies on the assessment of medical resource demands in regions or countries of different sizes.

Therefore, this study builds a transmission dynamics model based on scenario simulation, simulates the development trend of COVID-19 in four epidemic scenarios in cities of different sizes, and evaluates the demands of COVID-19 in cities with different population sizes and epidemic scenarios. Laboratory testing personnel, material resources, and protective materials provide a decision-making basis for the dynamic allocation of emergency materials and the allocation of human resources.

## 2. Materials and Methods

### 2.1. Concept Definition

The nucleic acid screening process resources consist of three parts: ① Human resources, including nucleic acid sampling personnel, sampling service auxiliary personnel, laboratory testing personnel, and laboratory-related auxiliary personnel. ② Laboratory testing resources, including nucleic acid extraction instruments (96 wells), fluorescent PCR amplifiers (96 wells), centrifuges, vortex mixers, micro-adjustable sample dispensers, and other instruments, as well as A2-type double biological safety cabinets, inactivation Incubators, ultra-clean workbenches, movable UV lamps, air sterilizers, and other necessary equipment for nucleic acid testing; ③ Protection equipment for staff, including N95 and above protective masks, protective clothing, isolation gowns, latex gloves, and waterproof boot covers.

This study defines a city with a population of 10 million as a large city, a city with a population of 5 million as a medium city, and a city with a population of 500,000 as a small city.

Non-pharmaceutical interventions (NPIs) are an important means to contain the development of the COVID-19 epidemic. The NPIs implemented in this study included mask-wearing, close contact tracing, travel restrictions, closure of schools and workplaces, cancellation of mass gatherings, and hand sanitization [24,25].

### 2.2. Model Building Assumptions

(1) There is currently no specific treatment drug for COVID-19, which can produce immune protection after vaccination.

(2) Asymptomatic infection is contagious, and its infectious index is the same as that of mild patients.

(3) Asymptomatic infected and confirmed patients are no longer contagious due to isolation and infection control measures.

(4) After patients enter the recovery period, they may develop symptoms that are no longer contagious.

(5) This study assumes cities with population sizes of 10 million, 5 million, and 500,000 with uniform population distribution.

### 2.3. Epidemiological Model

This study uses an improved SEIR model to simulate the spread of COVID-19 in cities with large, medium, and small populations in China under four scenarios. The improved SEIR framework takes into account different infection states and the widespread vaccination of the population in the traditional 4 compartments—susceptible state (S), exposed state (E), infected state (I), and recovered state (R) [26], the asymptomatic infected person’s room (A), the vaccinated susceptible person’s room (V), the isolation room (Q), and the hospitalization room (H) were added, and the infected person’s room (H) was added. Patients (I) were further divided into mild patients (I_1_) and severe/critical patients (I_2_) to become the SVEAI_i_QHR model. The architecture diagram of the propagation process of the COVID-19 SVEAI_i_QHR model is shown in Figure 1.

That is to say, in the total population, except for those who have been immunized after being vaccinated against COVID-19, the rest of the population is susceptible to COVID-19. After exposure, they are infected at a certain rate. The incubation period is contagious. Case diagnosis is delayed, so it is confirmed after a certain delay. Symptomatic patients are hospitalized according to the prevention and treatment plan after diagnosis, and asymptomatic and mildly infected patients are admitted to Fangcang shelter hospitals according to the current epidemic prevention and control requirements. Quarantine, and each at a different rate of removal (recovery/death). The specific differential equation is shown in Formula (1). The descriptions of important parameters in the SVEAI_i_QHR model and their sources are shown in Table 1.
(1)$$\{\begin{array}{c} \frac{\mathrm{dS}}{\mathrm{dt}}=-\frac{S}{N}\left[\left(A+I_{1}\right)\beta_{1}+I_{2}\beta_{2}\right]-\mathrm{Sw}\\ \frac{\mathrm{dE}}{\mathrm{dt}}=\left[\frac{S}{N}+\frac{V}{N}\left(1-\alpha \right)\right]\left[\left(A+I_{1}\right)\beta_{1}+I_{2}\beta_{2}\right]-\mathrm{Ete}\\ \frac{\mathrm{dV}}{\mathrm{dt}}=\mathrm{Sw}-\frac{V}{N}\left[\left(A+I_{1}\right)\beta_{1}+I_{2}\beta_{2}\right]\left(1-\alpha \right)\\ \frac{\mathrm{dA}}{\mathrm{dt}}={\mathrm{EteDf}}_{1}-{\mathrm{ADq}}_{1}\\ \frac{{\mathrm{dI}}_{1}}{\mathrm{dt}}={\mathrm{EteDf}}_{2}-I_{1}{\mathrm{Dq}}_{2}\\ \frac{{\mathrm{dI}}_{2}}{\mathrm{dt}}={\mathrm{EteDf}}_{3}-I_{2}{\mathrm{Dq}}_{3}\\ \frac{\mathrm{dQ}}{\mathrm{dt}}={\mathrm{ADq}}_{1}+I_{1}{\mathrm{Dq}}_{2}-{Q\gamma }_{1}\\ \frac{\mathrm{dH}}{\mathrm{dt}}=I_{2}{\mathrm{Dq}}_{3}-{H\gamma }_{2}\\ \frac{\mathrm{dR}}{\mathrm{dt}}={Q\gamma }_{1}+{H\gamma }_{2}\\ N=S+V+E+A+I_{1}+I_{2}+Q+H+R \end{array}$$

### 2.4. Scenario Construction

Based on the scenario construction theory, this study constructed four epidemic scenarios for large, medium, and small cities of different sizes through expert consultation and group discussion. In the construction of the scenarios, based on the characteristics of the current domestic clustered epidemics with multiple and frequent outbreaks, the construction of the scenarios in this study is based on the outbreak scenarios and is set according to whether nucleic acid screening measures are taken and the effect of NPIs implementation. The specific scenarios are as follows:

Scenario 1: There are many initial imported cases, asymptomatic infections account for 85%, COVID-19 vaccination is available, prevention and control measures are strictly implemented and effective (80%), nucleic acid screening measures are not implemented for all employees, and the initial input is quantitatively set There are 10 cases, including 7 asymptomatic infections, 2 mild patients, and 1 severe/critical patient (Table 2), all of which are contagious, with no deaths or cures.

Scenario 2: In a disease outbreak scenario, only nucleic acid testing is carried out for all employees, but no other prevention and control measures are implemented. It is assumed that in the early stage of this scenario, imported cases are the mainstay, various prevention and control measures are relaxed in the normalized epidemic prevention and control stage, and only routine (once in 7 days) nucleic acid screening is performed for all employees, and daily nucleic acid screening is performed after cases are found. Screening. Nucleic acid testing of all staff is based on a 1:10 mixed test, and other personnel (diagnosed COVID-19 patients, other medical treatment, hospitalized patients and accompanying staff, close contact, sub-close contact) are all tested at 1:1. Other details are set as in scenario 1.

Scenario 3: There is an initial outbreak and community transmission. Strict prevention and control measures are implemented, and cases and close contacts are strictly managed. The prevention and control measures are effective (80%), and nucleic acid testing is implemented for all staff. The initial case setting is the same as scenario 1. The implementation of nucleic acid testing is the same as scenario 2.

Scenario 4: The epidemic intensity is high, the prevention and control measures are effective (80%), and nucleic acid screening measures are implemented. However, according to the current third edition of China’s nucleic acid testing organization implementation guidelines, only regional nucleic acid screening measures are required after cases appear. Therefore, based on routine nucleic acid screening (once every 7 days), once a case is found, 10% of the local population will be organized to carry out regional nucleic acid screening, and further divided into closed control areas, control areas, and prevention areas.

### 2.5. Laboratory Testing Related Resource Parameters

The calculation indicators of laboratory testing personnel and materials-related resources are derived from the epidemic characteristics of COVID-19, literature reports, guidelines for the prevention and control of the COVID-19 epidemic, guidelines for the implementation of nucleic acid testing organizations, expert consultation results, and scenario assumptions. According to the above model, predict the peak number of current illnesses under the implementation of prevention and control measures in different scenarios; determine the number of asymptomatic infections, mild, severe/critical patients, and the number of outpatient clinic visits and their population in cities of different sizes The scope and amount of nucleic acid screening for routine and full staff. Number of people sampled (number of nucleic acid sampling personnel = number of population ÷ 360), sampling staff: service guarantee staff = 1:3. Calculate the ratio of sampling tubes to different detection resources, laboratory testers (10,000:24), laboratory-related auxiliary personnel (10,000:15), 96-well nucleic acid extraction instrument (10,000:4), 96-well fluorescent PCR amplifier (10,000:10), A2 type double biological safety cabinet (10,000:3), micro-adjustable sampler (10,000:4, 10,000:3), single-tube palm centrifuge, 8 tubes, 96-well plate centrifuge, small vortex Mixer (10,000:2), etc. (Appendix A). Calculate the number of protective equipment based on sampling, testing the peak number of relevant personnel, and personal protection standards.

The reference value of the existing laboratory testing manpower reserve in cities with a population of 10 million in China: an average of 1629 laboratory personnel in general hospitals, and an average of 195 CDC laboratory personnel [37]; the reference value of existing laboratory testing labor reserves in cities with a population of 5 million in China: The average number of inspection personnel in general hospitals is 814, and the average number of inspection personnel in CDC is 98 [37]; the reference value of the existing laboratory testing manpower reserve in cities with a population of 500,000 in China: the average number of inspection personnel in general hospitals is 81, and the average number of inspection personnel in CDC is 10 [37].

### 2.6. Model Validation

Once an epidemic occurs in mainland China, strict prevention and control measures will be taken immediately, so that the epidemic can be controlled in a short period, and the epidemic time is short, making it difficult for the model to accurately fit the epidemic data in mainland China. The fifth wave of the epidemic in Hong Kong has had a high prevalence for some time. The fifth wave of the epidemic lasted from January 2022 to April 2022. Given that the reported data of confirmed cases in Hong Kong are relatively complete compared with other areas in the same period [38], they are conducive to the spread of the disease. Parametric testing under natural propagation. Therefore, this study compared the predicted number of infected people simulated by the SVEAI_i_QHR model with the actual number of daily reported cases of the fifth wave of the epidemic in Hong Kong to verify the model. The mean absolute percent error (MAPE) = 1.67%, and the correlation coefficient r = 0.998. It shows that the model fitting effect is good, and the simulated curve is in good agreement with the Hong Kong epidemic data (Appendix A). Hong Kong Epidemic data comes from: https://ourworldindata.org/coronavirus(accessed on 10 August 2022).

## 3. Results

The SVEAI_i_QHR model is used to simulate the number of asymptomatic infections and confirmed patients in different scenarios in large, medium, and small cities, and estimates the nucleic acid testing resources and protective equipment demands based on the number of infections and the implementation of prevention and control measures.

### 3.1. Scenario 1

In large cities, for example, the epidemic was brought under control at about 150 days, when the number of infected people was about 100,000. The epidemic reached its peak in 114 days. At this time, the peak numbers of asymptomatic infected patients, mild patients, and severe/critical patients were 994,291, 149,144, and 13,470, respectively (Figure 2a). The 96-well nucleic acid extraction instrument, fluorescent PCR amplifier, A2 type double biological safety cabinet, micro-adjustable sampler, single-tube palm centrifuge, 96-well plate centrifuge, eight tubes, vortex mixer, eight channels The number of laboratory tests such as pipettes, inactivation incubators, and other related equipment ranged from 117 to 1169 (Figure 3a, Table 3). The peak cumulative consumption of N95 and above protective masks, protective clothing, and other protective equipment is 29,250 sets (Figure 4a). Similarly, in medium and small cities, the epidemic peaked at 110 days and 90 days, respectively; the total number of peak infections was 579,638 and 58,585, respectively (Figure 2, Appendix A). Laboratory testing-related equipment ranged from 59–586 and 6–59 sets, respectively, and the cumulative consumption of protective equipment was 14,654 and 1481 sets (Figure 3 and Figure 4, Table 3). Even if the epidemic prevention and control did not carry out nucleic acid testing for all staff in this scenario, the total daily maximum demand for nucleic acid sampling personnel in large, medium, and small cities and the maximum daily demand for laboratory testing personnel have both exceeded the current existing testing personnel, which are 3.7 times that of hospital testing personnel, respectively, CDC inspectors 31.0 times (Table 3).

### 3.2. Scenario 2

Taking a big city as an example, the epidemic is basically under control in about 80 days, and the number of infected people can reach 50,000 at this time. The epidemic peaked at 56 days. At this time, the peak numbers of asymptomatic patients, mild patients, and severe/critical patients were 472,547, 56,975, and 10,318, respectively (Figure 2b). The number of detection-related supporting equipment ranged from 636 to 6355 sets (Figure 3b, Table 3), and the peak cumulative consumption of disposable protective equipment was 226,495 sets (Figure 4a). Similarly, in this scenario, medium-sized cities, and small cities reach the peak of the epidemic in 54 and 45 days, respectively, with a peak population of 270,301 and 27,166 people (Figure 2, Appendix A), and laboratory testing-related equipment is 319−3187. The total consumption of protective equipment is 113,308 and 11,404 sets (Figure 3 and Figure 4, Table 3). Under this scenario, the demand for nucleic acid screening-related staff, laboratory testing equipment, and personal protective equipment has increased sharply, which is much higher than the amount required for nucleic acid testing of non-full staff. The total maximum daily demand for nucleic acid sampling personnel and laboratory testing personnel in large, medium, and small cities has exceeded that of hospital inspection personnel in large, medium, and small cities in my country by 26.4 times, and 220.7 times more than that of the CDC inspection personnel (Table 3).

### 3.3. Scenario 3

The total number of peak cases in this scenario will not exceed 50. Under this scenario, the epidemic situation in large, medium, and small cities peaked on the 6th, 6th, and 2nd days of the epidemic, with 50, 38, and 26 people at the peak, respectively (Figure 2, Appendix A). Large, medium, and small cities require 100–999, 50–499, and 5–49 sets of laboratory testing equipment, respectively; the consumption of disposable protective equipment is 191,420, 95,587, and 9335 sets, respectively (Figure 3 and Figure 4, Table 3). In this scenario, epidemic prevention and control are carried out for nucleic acid testing of all staff. The maximum daily demand for specimen sampling personnel is similar to that of scenario 2. However, due to the implementation of strict prevention and control measures, the number of cases has dropped significantly. Large, medium and small cities have the largest daily demand for nucleic acid sampling personnel and laboratory testing personnel. The total demand has exceeded 18.5 times the inspection personnel of hospitals in large, medium, and small cities in my country, and 154.5 times the inspection personnel of the CDC (Table 3).

### 3.4. Scenario 4

It can be seen from the above results that if nucleic acid screening of all employees is carried out after a case occurs, although nucleic acid-positive persons can be found in time if a positive person is found to take nucleic acid testing of all employees immediately, it is not in line with the current prevention and control measures. The “Regional Implementation Guidelines for Novel Coronavirus Nucleic Acid Testing Organizations (Third Edition)” [39] requires that after the outbreak of the novel coronavirus pneumonia, based on accurate and rapid flow investigation and community management and control, scientific research and judgment on the risk of epidemic transmission and delineation of regional nucleic acid testing Scope for nucleic acid screening. Therefore, in this scenario, after weekly routine testing and positive detection, daily regional nucleic acid screening is changed to delineate sealing and control areas, control areas, and prevention areas, and conduct nucleic acid screening for people in the designated areas, respectively. Among them, the people in the closed and controlled areas and the control areas are subjected to a single inspection, and the population in the prevention area is mixed and tested. Based on reducing costs, accurate prevention and control are carried out, and infected persons are detected in time. Under this scenario, the epidemic in large, medium, and small cities peaked on 17, 14, and 7 days, respectively, and the number of infected people at the peak was 74, 50, and 33, respectively (Figure 2, Appendix A); the number of laboratory testing-related equipment was 11–113, 6–62, 2–17 sets; the cumulative consumption of disposable protective equipment is 19,255, 9665, and 1036 sets (Figure 3 and Figure 4, Table 3). The total maximum daily demand for nucleic acid sampling personnel and laboratory testing personnel in large, medium, and small cities has exceeded 1.8 times that of hospital inspection personnel in large, medium, and small cities in my country, and 15.6 times more than the CDC inspection personnel (Table 3).

## 4. Discussion

This study aims to combine the prevention and control experience and laboratory resource requirements of China and other countries during the outbreak of COVID-19, focusing on the analysis of the impact of regular nucleic acid testing on epidemic prevention and control, and constructing scenarios based on the transmission dynamics model, and comprehensively analyze my country’s large The current situation of medical and health resource demand in small and medium-sized cities provides theoretical support and reference for emergency preparedness of medical resources and adjustment of prevention and control strategies under the COVID-19 pandemic and also provides guidance for resource input and allocation of prevention and control materials.

The comparison of the results of scenarios 2 and 3 shows that by increasing prevention and control measures, the number of cases can be greatly reduced, and the epidemic can be effectively controlled more quickly, compared with only taking nucleic acid testing of all staff. For example, wearing a mask, on one hand, can protect oneself from others, on the other hand, can prevent COVID-19-infected people from spreading to others. To reduce the total number of COVID-19 infections and deaths, and this effect will increase with the effectiveness of masks [40]; COVID-19 is mainly transmitted by droplets, whereas social distancing reduces droplet transmission in the population by reducing person-to-person contact [41]. A study from China also showed that social distancing measures could reduce the number of infections by 98% [42]. The research results also show that if NPIs are not implemented in the case of implementing nucleic acid screening for all employees, the demand for laboratory testing resources will increase by at least 6 times. It can be seen that strong prevention and control measures are an important means to significantly reduce the scale of the epidemic and reduce the resource load of the health system. Relevant studies have shown that the purpose of implementing NPIs is to reduce the spread of the epidemic, thereby delaying the peak and reducing the scale of the epidemic, buying more preparation time for the healthcare system, and enabling the potential of vaccines and drugs to be used later [43]. Stringent NPIs are feasible in the short term, but their long-term operability remains to be seen due to their long-term impact on socioeconomic costs [44]. While COVID-19 is an emerging infectious disease, the antiviral drugs used in modern medicine are of little use, and the medical resources to treat all cases are also limited. As a result, containing the COVID-19 epidemic has placed a huge burden on health systems and societies in many countries. From a public health perspective, countries should rationally plan NPIs implementation strategies and related resource allocations [45] to maximize the benefits of these interventions and minimize the health, social, and economic impacts of COVID-19 around the world [46].

The comparison of the results of scenarios 1, 2, and 3 showed that the combined effect of nucleic acid screening and NPIs intervention was higher than that of single nucleic acid screening or NPIs intervention. Studies have shown that a combined strategy of personal protection and quarantine testing is the best option from the perspective of effectiveness and cost-effectiveness in controlling the spread of COVID-19 [47]. Therefore, whether it is a large, medium, or small-scale city, especially a city with a medium or low level of medical resources, strict prevention and control strategies should be implemented in the early stage of the epidemic, and “early detection, early reporting, early isolation, and early treatment” must be adhered to. and other measures, implement travel restrictions, increase social distance and other strategies, and at the same time use the window period to allocate medical resources as much as possible, adjust prevention and control strategies and allocate medical resources promptly based on existing resources, to alleviate the high epidemic situation of Scenario 1 and Scenario 2. The phenomenon of a serious shortage of resources [48].

A comparison of the results of scenarios 3 and 4 suggests that regional nucleic acid screening can maintain the required manpower and laboratory testing resources at a lower level while reducing the number of cases. Relevant research guidelines [39] proposed that based on scientific research and judgment of the epidemic situation, the detection scope should be accurately delineated, and the nucleic acid detection should be changed from “all staff” to “regional”. The size of the area is determined by the need for epidemic prevention and control and guide localities to complete nucleic acid testing in the designated areas. The spread of novel coronavirus variants is highly concealed. Nucleic acid screening for key areas and key populations is of great significance to find potential sources of infection as soon as possible and cut off the chain of virus transmission as soon as possible. A Chinese study has shown that nucleic acid testing can identify positive cases early in an outbreak. Community-wide nucleic acid testing has shown unique advantages in rapidly screening large populations compared with hospital-based strategies [49]. Therefore, mass testing is one of the most effective interventions to control the epidemic. There are also studies that show that large-scale nucleic acid testing will indeed save a lot of laboratory resources and human resources compared to full-staff nucleic acid testing [50]; secondly, by carrying out large-scale nucleic acid testing supplemented by some prevention and control measures, it is indeed possible to avoid the cost of closing the city. 

There are still some limitations in this study. First, in terms of parameter selection of infectious disease models, most of the published literature is referred to. However, due to differences in model parameters in different regions and different periods, further verification by empirical data is lacking. Second, because this study focuses on the impact of nucleic acid screening measures on the development of the epidemic, the medical and health resources in this article only include laboratory testing manpower and material resources under the emergency prevention and control of COVID-19 study and do not include the relevant resources for epidemic prevention and control. The main clinical and epidemiological prevention and control resources can be further improved after the data in the future guidance documents are filled in. Therefore, follow-up in-depth research can be deeply analyzed from these aspects. Third, the cities involved in this study are cities with populations of 10 million, 5 million, and 500,000, which are representative to some extent; however, the research results may not be fully generalized to other regions.

Therefore, future research should try to use more epidemic data to adjust and optimize model parameters to ensure the accuracy of model simulation in the aspect of parameter selection of infectious disease models. More guidance documents on clinical and epidemiological resources need to be collected to improve the study of health resource requirements, not just laboratory testing resources, to comprehensively assess the resources required after an outbreak. In addition, resource simulation analysis should be carried out considering different population sizes and scenarios under different prevention and control conditions, to promote the extrapolation of the research results to different countries and regions.

## 5. Conclusions

This study found that for large, medium, and small cities, the use of NPIs and regional nucleic acid screening can contain the development of COVID-19, and alleviate the shortage of health resources to a certain extent. Against the backdrop of the global COVID-19 pandemic, there is still a shortage of professionals related to nucleic acid sampling and testing in China. Therefore, a manpower reserve of professional nucleic acid sampling and laboratory testing is urgently needed to implement effective nucleic acid screening measures in the event of an outbreak. In addition, this study not only provides a decision-making basis for the adjustment of prevention and control strategies and resource preparation during the COVID-19 epidemic but also provides ideas and methods for other countries and regions to assess the resource needs of nucleic acid screening. It is suggested that cities or regions with different population sizes and medical resources should flexibly implement prevention and control measures according to the specific situation after the outbreak of the epidemic, assess laboratory testing resources and human resources needs as soon as possible, and prepare and allocate materials and personnel.

## Figures and Tables

**Figure 1 ijerph-19-13343-f001:**
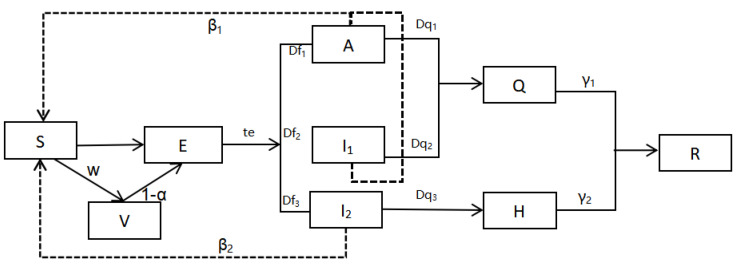
The structure of the SVEAI_i_QHR model of COVID-19.

**Figure 2 ijerph-19-13343-f002:**
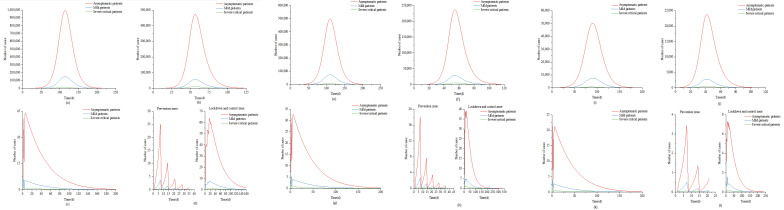
Epidemic development under scenarios 1–4 of cities of different scales. Panels (**a**–**d**) correspond to the change in the number of asymptomatic infected persons, mild patients, and severe/critical patients under the big city scenarios 1–4; panels (**e**–**h**) correspond to the asymptomatic infections under the medium city scenarios 1–4, the number of mild patients, and the number of severe/critical patients over time; panels (**i**–**l**) corresponds to the change in the number of asymptomatic infections, mild patients, and severe/critical patients in small city scenarios 1–4.

**Figure 3 ijerph-19-13343-f003:**
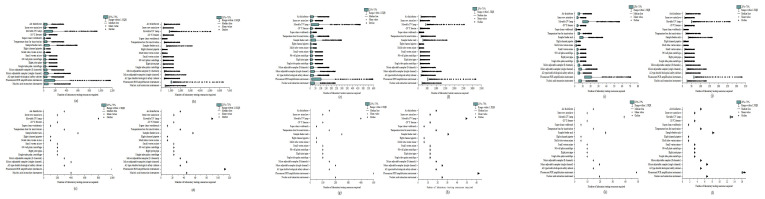
Laboratory testing resource requirements under different scale city scenarios 1–4; panels (**a**–**d**) correspond to laboratory testing resource requirements under large city scenarios 1–4; panels (**e**–**h**) correspond to medium city scenarios 1–4, demand for laboratory testing resources; panels (**i**–**l**) correspond to the demand for laboratory testing resources in small city scenarios 1–4.

**Figure 4 ijerph-19-13343-f004:**
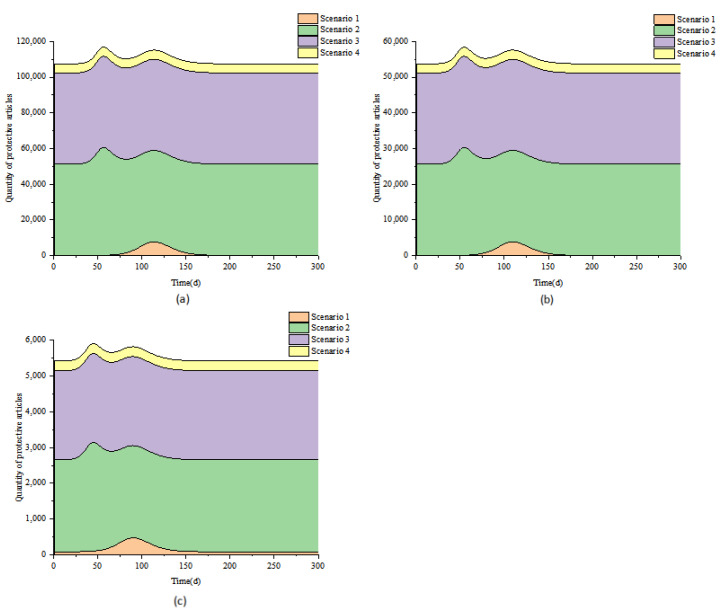
Time-dependent changes in demand for protective equipment under 1–4 under different scale city scenarios, (**a**) large cities; (**b**) medium cities; (**c**) small cities.

**Table 1 ijerph-19-13343-t001:** SVEAI_i_QHR model parameter meaning and value.

Parameter Name	Meaning	Model Value	Sources
w	Daily vaccination rate	0.005~0.015	Parameter estimation
α	Vaccine Protection Rate	0.14 (1 dose)	Reference [27]
ρ	Probability of an exposed person becoming an infected person	0.50	Reference [28]
β_1_	Infection index of asymptomatic infected persons and mild confirmed cases	4.46 (3.40~5.50)	References [29,30]
β_2_	Infection index of severe/critical confirmed cases	6.50 (5.00~8.00)	Reference [31]
γ1	The recovery rate of asymptomatic infections and mild confirmed cases	0.17	Reference [32]
γ2	The recovery rate of severe/critical confirmed cases	0.06	Reference [33]
te	The rate at which exposed persons progress to infected persons (reciprocal of incubation period)	1/4.40	Reference [34]
Df_1_	The proportion of asymptomatic infections	0.85	Reference [35]
Df_2_	The proportion of mild confirmed cases	0.1275	Reference [36]
Df_3_	The proportion of severe/critical confirmed cases	0.0225	Reference [36]
Dq_1_	Time from the discovery of asymptomatic infection to isolation(d)	1	Parameter estimation
Dq_2_	Time from the discovery of mild confirmed cases to their isolation(d)	1	Parameter estimation
Dq_3_	Time from the discovery of severe/critical confirmed cases to their hospitalization(d)	1	Parameter estimation
θ	An effective protection rate of masks	0.50 (0.50~0.85)	Reference [14]
a: Vaccination rate = Planned daily vaccinations/Total population			

**Table 2 ijerph-19-13343-t002:** Initial values of scenario simulation for cities of different sizes.

Meaning	Large City	Medium City	Small City
Total number of the model (N)	10,000,000	5,000,000	500,000
Initial susceptible population (S_0_)	1,999,630	999,730	99,830
Number of initial vaccinations (V_0_)	8,000,000	4,000,000	400,000
Number of initial exposure (E_0_)	360	260	160
The initial number of asymptomatic infections (A_0_)	7	7	7
The initial number of mild confirmed cases (I_1_)	2	2	2
The initial number of severe/critical confirmed cases (I_2_)	1	1	1

**Table 3 ijerph-19-13343-t003:** Maximum daily demands of various resources under different scale city scenarios 1–4.

Category	Scenario1 (Large City)	Scenario2 (Large City)	Scenario3 (Large City)	Scenario4 (Large City)	Scenario1 (Medium City)	Scenario2 (Medium City)	Scenario3 (Medium City)	Scenario4 (Medium City)	Scenario1 (Small City)	Scenario2 (Small City)	Scenario3 (Small City)	Scenario4 (Small City)
Maximum daily demand of staff												
Nucleic acid sampling person	3247	27,778	27,740	2778	1627	13,889	13,852	1389	164	1389	1353	139
sampling service auxiliary personnel	9742	83,333	83,219	8333	4881	41,667	41,555	4167	493	4167	4058	417
Laboratory testing personnel	2805	15,253	2397	271	1406	7649	1197	150	142	792	117	41
Laboratory-related auxiliary personnel	1754	9533	1498	170	879	4780	748	94	89	495	73	25
Maximum daily demand for laboratory testing resources												
Nucleic acid extraction instrument (96 wells)	468	2542	399	45	234	1275	199	25	24	132	19	7
Fluorescent PCR amplification instrument (96 wells)	1169	6355	999	113	586	3187	499	62	59	330	49	17
A2 type double biological safety cabinet	351	1907	300	34	176	956	150	19	18	99	15	5
Micro-adjustable sampler (single channel)	468	2542	399	45	234	1275	199	25	28	132	19	7
Micro-adjustable sampler (8 channels)	351	1907	300	34	176	956	150	19	18	99	15	5
Single-tube palm centrifuge	234	1271	200	23	117	637	100	12	12	66	10	3
Eight joint pipe	234	1271	200	23	117	637	100	12	12	66	10	3
96-well plate centrifuge	234	1271	200	23	117	637	100	12	12	66	10	3
Small vortex mixer	234	1271	200	23	117	637	100	12	12	66	10	3
Multi-tube vortex mixer	117	636	100	11	59	319	50	6	6	33	5	2
Eight channel pipette	117	636	100	11	59	319	50	6	6	33	5	2
Sample feeder rack	585	3178	499	57	293	1593	249	31	30	165	24	8
Temperature box for inactivation	351	1907	300	34	176	956	150	19	18	99	15	5
Super clean workbench	117	636	100	11	59	319	50	6	6	33	5	2
−20 °C freezer	234	1271	200	23	117	637	100	12	12	66	10	3
Movable UV lamp	935	5084	799	90	469	2550	399	50	47	264	39	14
Inner row autoclave	234	1271	200	23	117	637	100	12	12	66	10	3
Air disinfector	351	1907	300	34	176	956	150	19	18	99	15	5

## Data Availability

Not applicable.

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
