# Peer review of "Effect of Nucleic Acid Screening Measures on COVID-19 Transmission in Cities of Different Scales and Assessment of Related Testing Resource Demands—Evidence from China"

_ijerph, 2022, doi:10.3390/ijerph192013343_

Round 1
Reviewer 1 Report
Dear authors,
The topic of the article is very important. Congratulations on the study. Here there are some suggestions for improvement:
2 - Materials and Methods
2.2. Model Building Assumptions (page 3)
Can you explain this topic?
(4) Once the patient recovers from the infection, there will be no second infection.
3 - Results
You do not need to repeat the dates for each scenario. This information is described in 2.4 - Scenario Construction.
5 - Conclusions
I suggest you show the importance of this study with the main dates of your research.
Author Response
- 1. 2 - Materials and Methods
2.2. Model Building Assumptions (page 3)
Can you explain this topic?
(4) Once the patient recovers from the infection, there will be no second infection.
Response:We are very grateful to the reviewer for your comments.First of all, there is indeed a secondary infection of COVID-19 patients, but since this study uses the compartment model to simulate the spread of COVID-19, and the characteristic of the SEIR compartment model is that the research population will gradually flow in each compartment as a whole, and the population in this study will eventually flow to the R compartment, so this assumption is only to explain its possible state in the R compartment at last.Secondly, we have some ambiguity in language expression. Now we have modified 2.2. Model Building Assumptions (page 3, (4)) of the manuscript and marked it in red. It is modified as "(4) After COVID-19 patients enter the recovery period, they may have symptoms, but they are no longer infectious".
- 2.3 - Results
You do not need to repeat the dates for each scenario. This information is described in 2.4 - Scenario Construction.
Response:Thanks to the reviewer for the suggestion.We have deleted the repeated description of the manuscript results. See the manuscript results section (pages 5-6) for details.
3.5 - Conclusions
I suggest you show the importance of this study with the main dates of your research.
Response:Thanks to the reviewer for the suggestion.We have revised the conclusions according to your suggestions, see manuscript 5 - Conclusions.The revised conclusion is as follows“This study found that for large, medium, and small cities, the use of NPIs and regional nucleic acid screening can contain the development of COVID-19, and alleviate the shortage of health resources to a certain extent. Against the backdrop of the global COVID-19 pandemic, there is still a shortage of professionals related to nucleic acid sampling and testing in China. Therefore, a manpower reserve of professional nucleic acid sampling and laboratory testing is urgently needed to implement effective nucleic acid screening measures in the event of an outbreak. In addition, this study not only provides a decision-making basis for the adjustment of prevention and control strategies and resource preparation during the COVID-19 epidemic but also provides ideas and methods for other countries and regions to assess the resource needs of nucleic acid screening. It is suggested that cities or regions with different population sizes and medical resources should flexibly implement prevention and control measures according to the specific situation after the outbreak of the epidemic, assess laboratory testing resources and human resources needs as soon as possible, and prepare and allocate materials and personnel.”
Reviewer 2 Report
The study presents model of transmission of COVID 19. The manuscript presents a novel idea and may be employed as AI tool.
Author Response
Thank you very much for your recognition of our study. We will continue to improve our studies in the future.
Reviewer 3 Report
Refere to attached file

Author Response
1.Research title
Recommend to use the term "Effect" instead of Impact
Response:Thanks to the reviewer for the suggestion.According to your suggestion, we have changed the title to "Effect of Nucleic Acid Screening Measures on COVID-19 Transmission in Cities of Different Scales and Resources of Assessment of Related Testing -Evidence from China. Please see the part marked red in the title of the manuscript for details.
- Abstract
-The abstract should include components of background; method; results; conclusion, even in an
organized way. Its nor regarded in abstract as well.
Response:Thanks to the reviewer for the suggestion.According to your suggestion, we have modified the summary as follows: "Abstract: Background: The COVID-19 epidemic is in its epidemic period, and China is still facing the dual risks of import and domestic rebound. To better control the COVID-19 pandemic under the existing conditions , the focus of this study is to simulate the nucleic acid testing for different population size cities in China to influence the spread of the COVID-19, assess the situation under different scenarios, the demand of the laboratory testing personnel, material resources, for the implementation of the nucleic acid screening measures, emergency supplies, and the configuration of human resources to provide decision-making basis. Methods: According to the transmission characteristics of COVID-19 and the current prevention and control strategies in China, four epidemic scenarios were assumed. Based on the constructed SVEAIiQHR model, the number of people infected with COVID-19 in cities with populations of 10 million, 5 million, and 500,000 was analyzed and predicted under the four scenarios, and the demand for laboratory testing resources was evaluated respectively. Results: For large, medium, and small cities, whether full or regional nucleic acid screening can significantly reduce the epidemic prevention and control strategy of different scenarios laboratory testing resource demand difference is bigger, implement effective non-pharmaceutical interventions and regional nucleic acid screening measures to significantly reduce laboratory testing related resources demand, but will cause varying degrees of inspection staff shortages. Conclusion: There is still an urgent need for laboratory testing manpower in China to implement effective nucleic acid screening measures in the event of an outbreak.Cities or regions with different population sizes and levels of medical resources should flexibly implement prevention and control measures according to specific conditions after the outbreak, assess laboratory testing and human resource need as soon as possible, and prepare and allocate materials and personnel." See Manuscript Abstracts for details.
-Method in abstraction section is not well described. Please extend it more. Which cities?
Procedure? Sample?
Response:Thanks to the reviewer for the suggestion.We have enriched the methods section of the abstract as you suggested. As follows " Methods: According to the transmission characteristics of COVID-19 and the current prevention and control strategies in China, four epidemic scenarios were assumed. Based on the constructed SVEAIiQHR model, the number of people infected with COVID-19 in cities with populations of 10 million, 5 million, and 500,000 was analyzed and predicted under the four scenarios, and the demand for laboratory testing resources was evaluated respectively. "See the Methods section of the manuscript abstract for details.
-Key words should be arranged based on Mesh Standard.
Response:Thanks to the reviewer for the suggestion.We have modified the keywords as follows according to the Mesh Standard. "COVID-19;Nucleic acid screening;Dynamical model of infectious disease;Laboratory testing"
3.Introduction
-Which country, city, region is it referred to? Please kindly specify. “The global novel coronavirus
pneumonia (COVID-19, new crown pneumonia) pandemic that broke out at the end of 2019 has 600 million
confirmed cases and 6.45 million deaths”
Response:We are very grateful to the reviewer for your comments.The interpretation of this sentence has been revised and marked in red in the introduction section of the manuscript.The modification is as follows:“As of August 14, 2022, The global novel coronavirus pneumonia (COVID-19, new crown pneumonia) pandemic that broke out at the end of 2019 has 600 million confirmed cases and 6.45 million deaths worldwide.”
-You’ve stated that “At present, the epidemic situation is not optimistic.”. as you know, covid-19 has been accompanied by myriad of unknown consequences. Therefore, when you point out: at present” when exactly does it refer to. Pleas specify by exact date. Maybe, the date for your study conceptualization. Response:We are very grateful to the reviewer for your comments.the expression "At present, the epidemic situation is not optimistic" is slightly repetitive with the follow-up "makes epidemic prevention and control more difficult", so , relevant descriptions have been deleted from the Introduction part of the manuscript.
-better to mention in one paragraph with respect to the cost imposed by Covid-19 all over the world and especially China. It highlights the necessity of such a valuable work. You can use the following references for that:
Dergaa, Ismail, et al. "COVID-19 vaccination, herd immunity and the transition toward normalcy: challenges with the upcoming sports events." Annals of Applied Sport Science 9.3 (2021): 0-0.
Response:Thanks to the reviewer for the suggestion.We have revised the introduction section according to your suggestion and added a related description of "the cost imposed by Covid-19 all over the world". And quoted the recommended literature, adding the following content: “The COVID-19 epidemic has not only caused a huge impact on the global economic and social development but also brought a huge threat to the physical and mental health of the population [3-6]”See the red section marked in the first paragraph of the introduction of the manuscript for details.
-Given the fact that it has been more than 3yoears since the pandemic and more than one years as its global control (in terms of epediomology); what’s the research necessity? No specific reason was found in introduction. Please kindly clarify.
Response:We are very grateful to the reviewer for your comments.The necessity of this study is that although the COVID-19 pandemic has been a global control disease for more than a year, many researchers have conducted relevant studies on the containment measures of COVID-19, but few studies have analyzed the impact of nucleic acid screening measures on the occurrence and development of COVID-19. As one of the most important measures to curb the development of COVID-19, nucleic acid screening measures will require a lot of human and material resources once the epidemic occurs. However, it is unclear how much resources cities with different population sizes need to carry out such large-scale nucleic acid screening measures. Therefore, this study simulated the impact of nucleic acid screening measures on COVID-19 and assessed the needs of laboratory testing personnel and material resources in cities with different population sizes. It not only provides a decision-making basis for the adjustment of prevention and control strategies and resource preparation during the COVID-19 epidemic but also provides a reference for the demand and allocation of nucleic acid screening-related resources.
-Life style changes has been one fixes reason in all Covid-19 research which potentially affects the obtained
results. You can point out it in introduction (following references are recommended);
Additionally, you can recommend it for future studies
Response:We are very grateful to the reviewer for your comments.According to your suggestion, we have added a related description of the influence of lifestyle in the second paragraph of the Introduction. adding the following content: “It added the following: "To curb the epidemic, many countries have implemented non-pharmaceutical interventions (NPIs) such as lockdowns, mask-wearing, and social distancing [13], and the implementation of these NPIs has also changed people's lifestyles. Now, people often wear masks when they go out and social distancing in public places. And studies have shown that social distancing of 1 meter or more or the use of masks significantly reduces the incidence of COVID-19 infection[14].”Please refer to the part marked red in the manuscript for details.
4.Method:
-Is the quoted sentence grammatically ok? “Human resources, including nucleic acid sampling personnel,
sampling service support.
Response:We are very grateful to the reviewer for your comments.We have corrected the relevant grammar and marked it in red in the manuscript. The modifications are as follows: ① Human resources, including nucleic acid sampling personnel, sampling service auxiliary personnel, laboratory testing personnel, laboratory related auxiliary personnel.
-What was the reason for choosing the cities (population: 10.000.000 people, so on). As resected authors
know, sampling should be based on either a logical basis or equation (e.g., G-power, Morgan table, etc…)
Response:We are very grateful to the reviewer for your comments.Many thanks for the reviewer's comments. According to China's urban scale classification standard, cities are divided into the following categories by taking the urban permanent resident population as the statistical caliber: cities with an urban permanent resident population of less than 500,000 are small cities; cities with an urban permanent resident population more than 500,000 and less than 1 million are medium cities; A city with a permanent urban resident population of more than 1 million but less than 5 million is a big city; Cities with a permanent urban population of more than 10 million are megacities. In this study, the above city size classification criteria were combined according to the occurrence and frequency of the epidemic in different cities in China, and the population sizes of 10 million, 5 million, and 500,000 were selected for simulation, to cover the possible epidemic situation in China as much as possible.
-How do you justify the quoted claim in Model Building Assumptions? “Once the patient recovers from the
infection, there will be no second infection”. That’s obviously clear that many people have been infected
even in a short time after the first infection.
Response:We are very grateful to the reviewer for your comments.First of all, there is indeed a secondary infection of COVID-19 patients, but since this study uses the compartment model to simulate the spread of COVID-19, and the characteristic of the SEIR compartment model is that the research population will gradually flow in each compartment as a whole, and the population in this study will eventually flow to the R compartment, so this assumption is only to explain its possible state in the R compartment at last.Secondly, we have some ambiguity in language expression. Now we have modified 2.2. Model Building Assumptions (page 3, (4)) of the manuscript and marked it in red. It is modified as "(4) After COVID-19 patients enter the recovery period, they may have symptoms, but they are no longer infectious".
-If designing a conceptual framework or experimental model has been used for foresight objectives or
predict future health, shouldn’t you used foresight methods in designing scenarios for your research?
Response:We are very grateful to the reviewer for your comments.We believe that the scenario analysis method is forward-looking. Scenario analysis, also known as scenario planning, is to realize future scenarios through hypothesis, prediction, simulation, and other methods, describe various potential results in the future and predict the possible impact of various scenarios in the future. Based on scenario analysis, the characteristics, occurrence, and development rules of emergencies can be simulated and analyzed, to help emergency decision-makers to better judge the possible results and impacts of emergencies. Emergencies such as COVID-19 are often characterized by sudden onset, uncertainty, high degree of harm and widespread, and lack of historical experience for reference. Therefore, it is of great significance to simulate the possible results of emergencies in the future by using scenario analysis. Based on the epidemic dynamics model, this study used scenario simulation to predict and analyze COVID-19 transmission and nucleic acid testing-related resources under different prevention and control measures and nucleic acid screening measures, to provide a reference for the formulation and adjustment of subsequent prevention and control measures.
-What have been the criterion for the comparison of the results of scenarios? As you’ve collected data as
demonstrated in table 3; you would analyze the data using inferential statistics.
Response:We are very grateful to the reviewer for your comments.In this study, we compared the number of asymptomatic infected patients, mild cases, and severe cases at the peak of the epidemic under different scenarios as the outcome indicators of different scenarios. The larger the number of infected patients, the larger the scale of the epidemic. Furthermore, we calculated the testing resources required for laboratory testing based on the ratio between the number of testing people and testing resources in the COVID-19 epidemic prevention and control guidelines
5.Discussion
-You are expected to focus more on possible mechanisms involved in obtained results. Your justification
regarding the differences between scenarios…..
Response:Thanks to the reviewer for the suggestion.We have modified the discussion section according to your suggestion and added the analysis of differences between different scenarios. Added the following content:“The comparison of the results of scenario 2 and scenario 3 shows that,compared with full nucleic acid testing, increasing prevention and control measures can significantly reduce the number of cases and control the epidemic more quickly and effectively. For example, wearing a mask, on one hand, can protect oneself from others, on the other hand, can prevent COVID-19-infected people from spreading to others; To reduce the total number of COVID-19 infections and deaths, and this effect will increase with the effectiveness of masks [41]; COVID-19 is mainly transmitted by droplet, while social distancing reduces droplet transmission in the population by reducing person-to-person contact [42]. A study from China also showed that social distancing measures could reduce the number of infections by 98% [43].”
“Comparing the results of scenario 3 and scenario 4 showed that regional nucleic acid screening could reduce the number of cases.A Chinese study has shown that nucleic acid testing can identify positive cases early in an outbreak. Community-wide nucleic acid testing has shown unique advantages in rapidly screening large populations compared with hospital-based strategies [50].” Please refer to the discussion section marked red in the manuscript for details.
-One of your study limitation seems to be the no of cities involved in your study. As a result, you mayvreport as study limitation.
Response:Thanks to the reviewer for the suggestion.According to your suggestion, we have added the following content in the study limitation section:“Third, the cities involved in this study are 10 million, 5 million, and 500,000 population cities, which are representative to some extent, but the research results may not be fully generalized to other regions.” Please refer to the section marked red in the research limitations of the manuscript.
-Please kindly report dome research recommendations for future studies based in your study limitations.
Response:Thanks to the reviewer for the suggestion.We have added suggestions for future research according to your suggestions. The following content was added:“Therefore, future research should try to use more epidemic data to adjust and optimize model parameters to ensure the accuracy of model simulation in the aspect of parameter selection of infectious disease models. More guidance documents on clinical and epidemiological resources need to be collected to improve the study of health resource requirements, not just laboratory testing resources, to comprehensively assess the resources required after an outbreak. In addition, resource simulation analysis should be carried out considering different population sizes and scenarios under different prevention and control conditions, to promote the extrapolation of the research results to different countries and regions.”Please refer to the section marked red on page10 of the manuscript.